# Efficacy of Botulinum Type A Injection for the Treatment of Postherpetic Neuralgia and Pruritus Persisting for More Than Four Years—A Case Report

**DOI:** 10.3390/medicina60081317

**Published:** 2024-08-14

**Authors:** Jihyun Song, Sang Sik Choi, Seok Jun Choi, Chung Hun Lee

**Affiliations:** Department of Anesthesiology and Pain Medicine, Korea University Medical Center, Guro Hospital, 148 Gurodong Road, Guro-gu, Seoul 08308, Republic of Korea; songjh0814@gmail.com (J.S.); clonidine@empal.com (S.S.C.); ijchei@naver.com (S.J.C.)

**Keywords:** postherpetic neuralgia, postherpetic pruritus, botulinum toxin type A, ptosis

## Abstract

*Background*: Postherpetic neuralgia (PHN) and postherpetic pruritus (PHP) are common complications of shingles that affect patients’ quality of life. PHN and PHP can be managed using various medications and interventional procedures; however, complications persisting for at least six months may hamper recovery. Subcutaneous injections of botulinum toxin type A (BTX-A) can control persistent PHN and PHP. *Case presentation*: A 71-year-old man presented at our hospital with itching and pain. He had been diagnosed with shingles in the ophthalmic branch of the trigeminal nerve one year previously. As the pain and itching persisted despite medication, a supraorbital nerve block, Gasserian ganglion block, epidural nerve block, and radiofrequency thermocoagulation were performed. A subcutaneous injection of BTX-A was administered into the ophthalmic area of the trigeminal nerve three years after the initial presentation. A decrease of >80% in pain and itching was reported after the injection; however, the left eyelid drooped and the eyeball shifted downward and outward immediately after the injection. No deterioration in vision or pupil dilation was observed, and almost complete resolution of these symptoms occurred spontaneously three months after the injection. Pain and itching continued to improve without further side-effects until six months after the injection. *Conclusions*: The subcutaneous injection of BTX-A may be an alternative treatment option for chronic and refractory neurological diseases such as PHN and PHP, which persist for four years and are resistant to conventional treatments. Nevertheless, care must be taken to minimize the risk of ptosis.

## 1. Introduction

Herpes zoster (HZ) is a disease caused by the reactivation of the varicella zoster virus (VZV), which lies dormant in sensory neurons and is characterized by a painful erythematous rash on the affected dermatome. The most distressing symptom in patients with shingles is pain, with postherpetic neuralgia (PHN) being the most distressing. Postherpetic neuralgia is neuropathic pain that persists for at least 90 days after the rash [1]. Insufficient pain management impairs the physical, emotional, and social well-being of patients, resulting in a reduction in their quality of life [1,2,3]. Postherpetic pruritus (PHP) is a complication of HZ that presents as chronic refractory pruritus in the affected areas. Although PHP is a rare complication of shingles, it is reportedly more common in patients with PHN of the trigeminal nerve than in other patients [4]. As with PHN, PHP affects the quality of life of patients [4,5]. 

The targets of the treatment strategies for PHN and PHP are symptomatic relief and improvement in the quality of life of patients. The pharmaceutical approach involves the administration of neuropathic drugs, such as tricyclic antidepressants, anticonvulsants, lidocaine patches, capsaicin, and narcotic analgesics [1,6,7]. In addition, interventional procedures, such as peripheral nerve block, epidural nerve block, sympathetic ganglion block, radiofrequency (RF) thermocoagulation, transcutaneous electrical nerve stimulation, and spinal cord stimulation, may be considered [1,6,7]. However, these treatment strategies have demonstrated limited efficacy in the treatment of patients with pain persisting for ≥6 months after the onset of HZ. This may be attributed to irreversible changes in the affected neural pathways [8]. In particular, the application of invasive treatment strategies, such as spinal cord stimulation or percutaneous nerve stimulation, may be limited for PHN and PHP arising in the trigeminal ganglion.

A subcutaneous injection of botulinum toxin type A (BTX-A) has been effective in the management of PHN [9,10,11,12]. However, most previous studies enrolled patients who had been affected by HZ for less than two years. To the best of our knowledge, no previous study has reported the therapeutic effect of BTX-A in patients with PHN and PHP persisting for four years after the onset of HZ infection. 

This is a case of PHN and PHP that persisted for four years despite pharmacological treatment and interventional procedures. The subcutaneous injection of BTX-A reduced the severity of PHN and PHP. 

## 2. Case Presentation

This study was approved by the Institutional Review Board of the Korea University Medical Center, Guro Hospital, Seoul, Republic of Korea (2023GR0235), on 7 June 2023. 

A 71-year-old man presented at our hospital with pain and itching. The patient was diagnosed as having HZ of the ophthalmic branch of the trigeminal nerve. He had reportedly received drugs and nerve block treatment at another hospital one year earlier. Nevertheless, the patient complained of persistent pain and itching, along with sleep disturbances. The patient was administered 150 mg of pregabalin twice daily, 5 mg of amitriptyline at night, and acetaminophen/codeine phosphate hydrate/ibuprofen twice daily at the time of presentation. However, his pain score measured using a numeric rating scale (NRS) was approximately 8/10 in the frontal, parietal, and temporal areas above the left eye. In addition, allodynia and severe itching were observed even in the absence of stimulation. Compared with the unaffected side, his ability to discriminate between pressure, temperature, and sense vibrations was reduced. No differences were observed in the appearance of forehead wrinkles between the affected and unaffected sides.

We focused on PHN and PHP rather than differential diagnoses because the patient’s symptoms had persisted for a year before presentation and he complained of pain and itching, beginning with a rash, in the skin areas typical of PHN and PHP. The creatinine levels and creatinine clearance levels were 1.08 mg/dL and 62 mL/min, respectively, indicating no impairment of renal function. Therefore, 75 mg of pregabalin was additionally administered twice daily, and a supraorbital nerve block was performed several times using 0.20% ropivacaine. Partial relief from pain and itching was reported 24 h after the peripheral nerve block was performed; however, the symptoms recurred. Increasing the pregabalin dose did not result in any significant improvement. The administration of acetaminophen/codeine phosphate hydrate/ibuprofen was discontinued, and the administration of 5 mg of oxycodone twice daily was initiated. In addition, a left Gasserian ganglion block (GGB) was performed with 0.8 mL of 0.75% ropivacaine and 0.2 mL of dexamethasone; however, no significant improvement was observed.

Because temporary symptomatic relief was achieved after the supraorbital nerve block, RF thermocoagulation of the supraorbital nerve was performed one year after presentation (two years after the development of shingles). Reductions in pain scores (NRS score 5/10) and itching (reduced by half) were observed after RF thermocoagulation. However, a recurrence of pain (NRS 7/10) and itching in the affected area was observed three months after RF thermocoagulation. Although a cervical epidural block was performed in the left cervical two-thirds, this did not result in any improvement. The patient’s pain and itching persisted despite more than a dozen supraorbital nerve blocks, two GGBs, two cervical epidural blocks, and one RF thermocoagulation.

After consultation with the patient, a subcutaneous injection of BTX-A was administered three years after the initial presentation (four years after the onset of the HZ infection). The procedure was as follows: the patient was placed in the supine position and vital signs were monitored throughout the procedure. A sterile dressing was applied to treated areas. BTX-A (150 IU; Nabota^®^, Daewoong Pharmaceutical, Seoul, Republic of Korea) was mixed with 5 mL of 2% lidocaine and 15 mL of saline. Subsequently, 0.5 mL (3.75 IU) per site, or a total of 20 mL (150 IU) of the solution, was injected into the affected area. The treatment area was determined by having the patient indicate all areas where he felt pain and itching, marking the skin area before the procedure. Subcutaneous BTX-A injections were then performed at 1.5 cm intervals within the marked skin segments (Figure 1).

The patient reported considerable improvement in pain and itching one week after receiving the injection; however, drooping of the left eyelid and downward and outward deviations in the eyeball were observed. No deterioration in vision or pupil dilation was observed and the patient did not report any significant inconvenience due to these complications (Figure 2).

Pain and itching continued to improve one month after the injection, and the administration of oxycodone was discontinued. An 80% reduction in pain and itching was reported three months after the subcutaneous injection of BTX-A had been administered. Consequently, the dose of pregabalin was reduced to 300 mg/day. The drooping eyelid and downward and outward deviations in the eyeball resolved spontaneously at the three-month time point. The patient continued to report improvements in pain and itching over the next three months. No further complications were observed. 

## 3. Discussion

This report describes a case in which a subcutaneous injection of BTX-A resulted in relief from pain and itching in a patient with PHN and PHP that had persisted for more than four years after HZ infection despite receiving medication and undergoing peripheral nerve block, epidural nerve block, GGB, and RF thermocoagulation.

Damage to the central and peripheral nerves caused by HZ affects the ability to suppress nociceptive signals. This lowers the threshold of nociceptors and results in spontaneous ectopic discharge. The resulting peripheral nerve cell death and central nervous system changes induce abnormal reorganization of the impulse transmission system and a disordered nerve distribution pattern, manifesting as spontaneous pain that progresses to PHN [1,13]. Well-established PHN, defined as the persistence of pain for at least six months, induces irreversible changes in the damaged ganglion. These changes hinder recovery [8]. The spontaneous firing of pruritic neurons in the central nervous system, an imbalance between the excitability and inhibition of secondary sensory neurons, and the selective preservation of peripheral pruritic nerve fibers from adjacent dermatomes unaffected by HZ can result in the occurrence of PHP [4,5]. PHN and PHP affected the trigeminal nerve more frequently than the other sites [4,5,14]. 

In this case, HZ affected the ophthalmic branch of the trigeminal nerve. The patient presented with well-established PHN and PHP during the first visit to our hospital. This limits the effectiveness of the initial treatment strategies of increasing the dose of pregabalin, administering narcotic analgesics, and performing peripheral nerve block, GGB, cervical epidural block, and RF thermocoagulation of the peripheral nerve branch.

The subcutaneous injection of BTX-A induces significant pain relief in patients with PHN and is relatively less invasive, and no serious complications have been reported to date. Therefore, BTX-A was subcutaneously administered. BTX-A reduces axonal transport by inhibiting the secretion of pain mediators (such as substance P, glutamate, and calcitonin-gene-related proteins) in the nerve endings and dorsal root ganglion, reducing inflammation around the nerve endings, and inactivating sodium channels [15]. Thus, BTX-A was used for the management of PHN based on this evidence and the findings of several prospective studies [9,10,11,12,15]. 

BTX-A alleviates pruritus by blocking the release of acetylcholine, a neurotransmitter that mediates chronic pruritus in presynaptic vesicles, via the inactivation of soluble N-ethylmaleimide-sensitive factor attachment protein receptors [16]. Furthermore, BTX-A reduced the transmission of itch signals in the C-fibers, suggesting that it may relieve PHP [16]. 

The subcutaneous injection of BTX-A had a significant effect on PHN and PHP in the current case, wherein PHP and PHN persisted for nearly four years. The administration of narcotic analgesics was discontinued one month after the injection, and the dose of neuropathic pain management drugs was tapered over the course of three months owing to the symptomatic relief achieved. 

Apalla et al. [9] reported that administering 100 IU of BTX-A resulted in a significant reduction in NRS pain scores for up to 16 weeks in patients with PHN. However, the treatment effect gradually decreased to <50% of the initial effect after 16 weeks. Hu et al. [10] reported that the efficacy period of 50–100 IU of BTX-A treatment ranges from one to two weeks. In contrast, Xiao et al. [11] reported that treatment with 200 IU BTX-A lasted for up to six months.

The treatment effect was maintained for more than six months in the current case, as 150 IU of BTX-A was used in narrow locations (frontal, temporal, and parietal), unlike in previous studies wherein BTX-A was injected in the torso. This finding is consistent with that of a study by Ri et al. [17], which suggested that the duration of the treatment effect correlated with the dose of BTX-A administered per injection site. Thus, the use of BTX-A at insufficient doses may shorten its efficacy period, necessitating repeated administration. This may induce the formation of anti-toxic antibodies, limit their clinical effects, and result in a significant loss of clinical efficacy. Therefore, a sufficient amount of BTX-A must be used during treatment to prevent the loss of clinical efficacy [9,18].

The risk-to-benefit ratio of BTX-A is low, with no serious safety concerns [9,10,11,19]. However, our patient developed ptosis at the site of BTX-A injection. Ptosis, a potential side-effect of the subcutaneous injection of BTX-A into the ophthalmic area of the trigeminal nerve, occurs in approximately 20% of cases [20,21,22]. This drooping of the eyelid may be attributed to the inadvertent paralysis of the levator palpebrae superioris caused by BTX-A [22]. Our patient developed ptosis and downward and outward deviations in the eyeball caused by muscle action. However, there were no abnormalities in the oculomotor nerves, and there was no impairment in vision or pupil dilation. These symptoms gradually improved and resolved almost completely three months after the injection. The site of BTX-A injection must be adjusted to prevent this complication [23]. Injecting BTX-A at titrated doses into the lower 60% of the frontalis muscle is recommended, because it is associated with eyebrow movement [24]. Deep injections may exacerbate the muscle-paralyzing effects of BTX-A. The injection of BTX-A into the superficial layer of the muscle rather than directly into the muscle or the periosteum beneath the muscle may result in less muscle paralysis [24]. In addition, ultrasound-guided BTX-A injections can more accurately control the depth of the needle, which may reduce complications, such as ptosis and muscle weakness [25]. Furthermore, the needle should be directed toward the parietal region to prevent the downward injection of BTX-A into the nerves of the muscles that raise the eyelids [26]. 

Sota et al. used 0.5% apraclonidine eye drops to treat ptosis after BTX-A injection into the forehead, orbicularis oculi, and corrugator muscles for migraine treatment [26]. Apraclonidine, an alpha-adrenergic receptor agonist, reverses ptosis by directly stimulating the sympathetic innervation of the superior tarsal muscles. In our patient, ptosis recovered spontaneously over time and resolved almost completely three months after injection. However, the use of 0.5% apraclonidine eye drops may be considered if ptosis does not recover naturally or if the patient experiences discomfort due to this complication.

This study had several limitations. First, the follow-up duration after the subcutaneous injection of BTX-A was only six months. Therefore, further studies with longer follow-up periods are warranted. Second, the emotional effects or incidence of sleep disorders were not considered when evaluating treatment effectiveness. Treatment effectiveness was determined based solely on the degree of pain and pruritus experienced by the patients. A large-scale randomized controlled trial assessing pain, pruritus, emotional functioning, and sleep disorders should be conducted to comprehensively evaluate the efficacy of subcutaneously injected BTX-A for the management of PHN and PHP.

## 4. Conclusions

The subcutaneous injection of BTX-A is a potential alternative for the management of persistent PHN and PHP that are resistant to conventional treatments. However, the subcutaneous injection of BTX-A into the ophthalmic area of the trigeminal nerve can result in ptosis; therefore, clinicians should exercise caution. Further prospective studies involving a large number of patients with long-term follow-up and different doses and injection techniques should be conducted to determine the efficacy and safety of this approach.

## Figures and Tables

**Figure 1 medicina-60-01317-f001:**
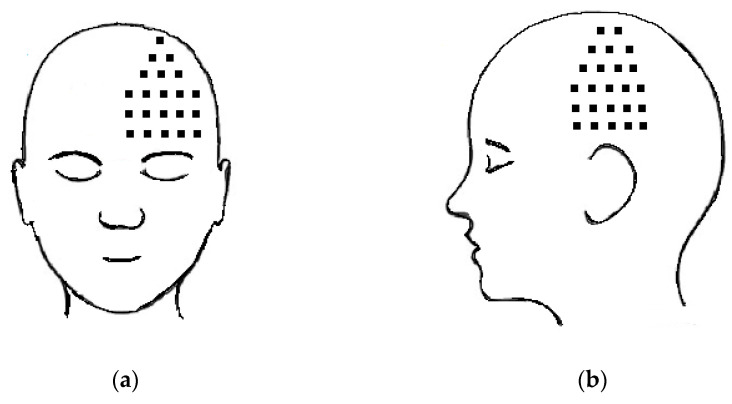
Locations of subcutaneous injections of botulinum toxin type A: (**a**) around the eyes and on the parietal forehead, and (**b**) at the temporal lobe.

**Figure 2 medicina-60-01317-f002:**
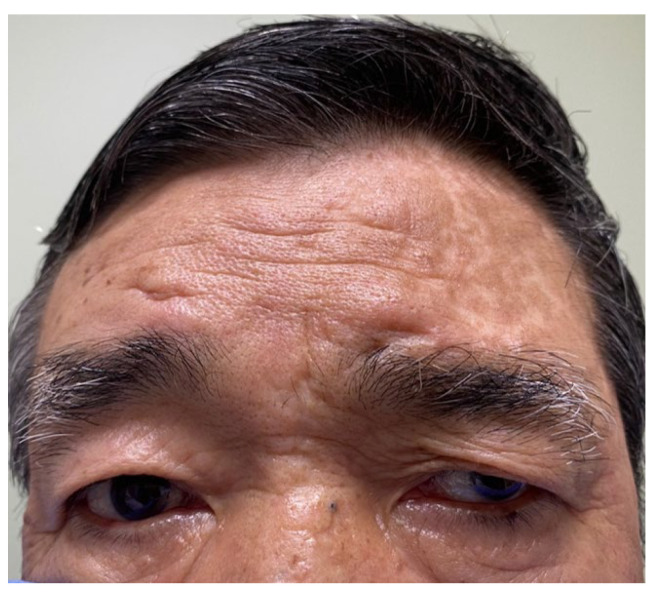
Effects of injecting botulinum toxin type A into the forehead above the left eye. Drooping of the left eyelid and downward and outward deviation in the eyeball can be observed.

## Data Availability

All relevant data are contained within this manuscript.

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
