# Peer review of "Efficacy of Botulinum Type A Injection for the Treatment of Postherpetic Neuralgia and Pruritus Persisting for More Than Four Years—A Case Report"

_medicina, 2024, doi:10.3390/medicina60081317_

Round 1

Reviewer 1 Report

Comments and Suggestions for Authors

General

1.       Describe and emphasize the novelty of the case report more clearly.

Introduction

1. Provide a brief background on the varicella-zoster virus and the typical progression to PHN (Lines 29-31).

2. Clarify the relationship or similarity between PHN to PHP, in more details (Lines 31-33).

3. Clarify the point about the limited efficacy of current treatments for persistent cases with more examples (Lines 42-45).

Case Presentation

1. Consider adding more details about the initial severity and response to earlier treatments (Lines 58-69).

2. Include results from additional diagnostic tests (e.g., MRI, CT scan) about the patient's condition (Lines 65-69).

3. Include more interventional treatment details about the patient's response and side effects.

4. Possibly include a follow-up plan or recommendations for managing complications.

5. Possibly add levels of inflammatory markers (CRP, ESR, Cytokines) to evaluate the systemic response to treatments.

Comments on the Quality of English Language

Minor language editing is needed.

Author Response

MEDICINA-3136866

Efficacy of botulinum type A injection for the treatment of postherpetic neuralgia and pruritus persisting for more than four years - A case report

Medicina

Response to reviewers’ comments

We appreciate the thoughtful comments and suggestions made by the reviewers. We have revised the manuscript based on these recommendations and believe that these changes have improved the quality of our manuscript. We have also provided point-by-point responses and descriptions of these changes.

We thank the reviewers and editor for their time.

Sincerely,

Chung Hun Lee

Department of Anesthesiology and Pain Medicine

Korea University Medical Center, Guro Hospital

Gurodong Road 148, Guro-Gu, Seoul 08308, Republic of Korea

Response to Reviewer #1

General

Describe and emphasize the novelty of the case report more clearly.

  • Thank you for your valuable suggestion. We have revised the text to clarify and emphasize the novelty of this case report

Introduction

  1. Provide a brief background on the varicella-zoster virus and the typical progression to PHN (Lines 29-31).
  • Thank you for your suggestion. In accordance, we have added a brief background on herpes zoster, varicella zoster virus, and the progression to postherpetic neuralgia at the beginning of the Introduction.
  • “Herpes zoster (HZ) is a disease caused by reactivation of the varicella zoster virus (VZV), which lies dormant in sensory neurons, and is characterized by a painful erythematous rash on the affected dermatome. The most distressing symptom in patients with shingles is pain, with postherpetic neuralgia (PHN) being the most distressing complication.”

  1. Clarify the relationship or similarity between PHN to PHP, in more details (Lines 31-33).
  • Thank you for your valuable suggestion. We have added the following sentences to the Introduction to clarify the relationship and similarities between PHN and PHP:
  • “Although PHP is a rare complication of shingles, it is reportedly more common in patients with PHN of the trigeminal nerve than in other patients [4].”

  1. Clarify the point about the limited efficacy of current treatments for persistent cases with more examples (Lines 42-45).
  • Thank you for this suggestion. We have added reasons for the limited efficacy or implementation of other interventions in cases like the current one in the Introduction.
  • “In particular, the application of invasive treatment strategies, such as spinal cord stimulation or percutaneous nerve stimulation, may be more limited for PHN and PHP arising in the trigeminal ganglion.”

Case Presentation

  1. Consider adding more details about the initial severity and response to earlier treatments (Lines 58-69).
  • Thank you for this comment. As suggested, we have described the patient's initial symptom severity and response to previous treatments in more detail in the Case Presentation.
  • “Nevertheless, the patient complained of persistent pain and itching, along with sleep disturbance due to the above symptoms.”

  1. Include results from additional diagnostic tests (e.g., MRI, CT scan) about the patient's condition (Lines 65-69).
  • We agree with the reviewer that additional diagnostic tests may be required in patients with long-term symptoms. However, we focused on diagnosing and treating the patient's symptoms, which had persisted for a year and were typical of PHP and PHN, rather than diagnosing other causes. Therefore, we did not perform additional diagnostic tests other than those required for the procedure (e.g., MRI and CT scans). We have added this information to the Case Presentation.
  • “We focused on PHN and PHP rather than the differential diagnoses because the patient's symptoms had persisted for a year before presentation and he complained of pain and itching, beginning with a rash, in skin areas typical of PHN and PHP.”

  1. Include more interventional treatment details about the patient's response and side effects.
  • Thank you for your comment. We have described the details of the interventional treatment and its response in more detail in the revised manuscript.
  • “The patient's pain and itching persisted despite more than a dozen supraorbital nerve blocks, two GGBs, two cervical epidural blocks, and one RF thermocoagulation.”
  • “Subsequently, 0.5 mL (3.75 IU) per site or a total of 20 mL (150 IU) of the solution was injected into the affected area. The treatment area was determined by having the patient indicate all areas where he felt pain and itching and marking the skin area before the procedure. Subcutaneous BTX-A injections were then performed at 1.5 centimeter intervals within the marked skin segments (Figure 1).”

  1. Possibly include a follow-up plan or recommendations for managing complications.
  • Thank you for your comment. We have added recommendations for the management of complications (such as ptosis) of BTX-A injection to the Discussion.
  • “Deep injections may exacerbate the muscle-paralyzing effects of BTX-A. Injection of BTX-A into the superficial layer of the muscle, rather than directly into the muscle or periosteum beneath the muscle, may result in less muscle paralysis [24]. Furthermore, the needle should be directed toward the parietal region to prevent the downward injection of BTX-A into the nerves of the muscles that raise the eyelids [25].
  • Sota et al. used 0.5% apraclonidine eye drops to treat ptosis occurring after BTX-A injection into the forehead, orbicularis oculi, and corrugator muscles for migraine treatment [25]. Apraclonidine, an alpha-adrenergic receptor agonist, reverses ptosis by directly stimulating the sympathetic innervation of the superior tarsal muscles. In our patient, ptosis recovered spontaneously over time and resolved almost completely three months after the injection. However, the use of 0.5% apraclonidine eye drops may be considered if ptosis does not recover naturally, or if the patient experiences discomfort due to this complication.”

  1. Possibly add levels of inflammatory markers (CRP, ESR, Cytokines) to evaluate the systemic response to treatments.
  • Thank you for your comment. We conduct laboratory investigations periodically, at least every six months, for patients taking long-term medications. We perform ESR, CRP, and other tests to check for infection at the procedure site before and after the intervention. However, there were no significant changes in the ESR or CRP levels in this patient before and after the procedure. Unfortunately, we were unable to test cytokine levels in this patient.

References

  1. Park, C.; John, H.; Lee, J.; Hong, S.; Kim, M.; Park, S.; Kim, J.H. The relative frequency of pruritus in postherpetic neuralgia patients presenting to the pain clinic and associative factors. Med (Baltim) 2022, 101, e30208. DOI:10.1097/MD.0000000000030208.
  2. Sandre, M. New anatomical insights into preventing brow ptosis with botulinum toxin-A use. Can Dermatol Today 2021, 2, 30–34.
  3. Omoigui, S.; Irene, S. Treatment of ptosis as a complication of botulinum toxin injection. Pain Med 2005, 6, 149–151. DOI:10.1111/j.1526-4637.2005.05029.x.

Reviewer 2 Report

Comments and Suggestions for Authors

In the case description, you mention that you injected 3.75 IU, but in the Discussion, you write that you administered 150 IU. Which is correct?

You provide a drawing of the area treated, but not how you decided on the area. Did you test the area affected (e.g. by testing sensation and defining borders, or did you just use the known innervation area of the nerve)? I suggest that the innervation area also be given, if you wish even in the same drawing to see how the areas overlap. In any case, the paper would be more helpful to clinicians if the following points were made clear: How did you decide on how many units to use, and how did you define borders for the treatment area?

You write that deeper injections ensure fewer side effects of paralysis, but then that '...injecting BTX-A into the superficial layer of the muscle reduces muscle paralysis compared...' Is this correct? What you are saying is that deeper injections lead to less paralysis and that superficial injections lead to less paralysis. Which is correct?

Author Response

MEDICINA-3136866

Efficacy of botulinum type A injection for the treatment of postherpetic neuralgia and pruritus persisting for more than four years - A case report

Medicina

Response to reviewers’ comments

We appreciate the thoughtful comments and suggestions made by the reviewers. We have revised the manuscript based on these recommendations and believe that these changes have improved the quality of our manuscript. We have also provided point-by-point responses and descriptions of these changes.

We thank the reviewers and editor for their time.

Sincerely,

Chung Hun Lee

Department of Anesthesiology and Pain Medicine

Korea University Medical Center, Guro Hospital

Gurodong Road 148, Guro-Gu, Seoul 08308, Republic of Korea

Response to Reviewer #2

In the case description, you mention that you injected 3.75 IU, but in the Discussion, you write that you administered 150 IU. Which is correct?

-> Thank you for your valuable comment. We injected a total of 20 mL (150 IU) of solution into the affected area, 0.5 mL (3.75 IU) per site. We have modified the text in the Case Description to clarify this.

-> “Subsequently, 0.5 mL (3.75 IU) per site or a total of 20 mL (150 IU) of the solution was injected into the affected area.”

You provide a drawing of the area treated, but not how you decided on the area. Did you test the area affected (e.g. by testing sensation and defining borders, or did you just use the known innervation area of the nerve)? I suggest that the innervation area also be given, if you wish even in the same drawing to see how the areas overlap. In any case, the paper would be more helpful to clinicians if the following points were made clear: How did you decide on how many units to use, and how did you define borders for the treatment area?

-> Thank you for your comment. However, when we decided on the treatment area, we focused on the location where the patient felt pain and itching rather than on the sensory test or innervation area. In other words, since the purpose of this treatment was to relieve the patient's pain and itching, we consulted the patient before treatment and asked him to identify all the areas where he felt pain and itching. We then marked all areas of the patient's skin before the procedure. Subsequently, we performed subcutaneous BTX-A injections at 1.5-centimeter intervals within the marked skin segments. We have added the above information to the text so that it will be helpful to those who read this paper.

-> The dose of BTX-A in previous studies varied from 50 to 200 IU [9-11]. Moreover, the patient had already borne exceedingly high expenses over a 4-year period of medication and various interventions before receiving the above treatment. Therefore, we decided to use the entire 1 ampoule (150 IU) of commercially available BTX-A (Nabota®, Daewoong Pharmaceutical, Seoul, Republic of Korea) for the patient to maximize the therapeutic effect for the cost borne. Thank you.

You write that deeper injections ensure fewer side effects of paralysis, but then that '...injecting BTX-A into the superficial layer of the muscle reduces muscle paralysis compared...' Is this correct? What you are saying is that deeper injections lead to less paralysis and that superficial injections lead to less paralysis. Which is correct?

-> We apologize for the confusion. Thank you for pointing this out. We hypothesized that injecting BTX-A into the superficial layer of the muscle, rather than directly into the muscle or periosteum beneath the muscle, could reduce muscle paralysis caused by BTX-A. We have revised the text to convey this information more accurately.

-> “Deep injections may exacerbate the muscle-paralyzing effects of BTX-A. Injection of BTX-A into the superficial layer of the muscle, rather than directly into the muscle or periosteum beneath the muscle, may result in less muscle paralysis [24]”

References

  1. Apalla, Z.; Sotiriou, E.; Lallas, A.; Lazaridou, E.; Ioannides, D. Botulinum toxin A in postherpetic neuralgia: A parallel, randomized, double-blind, single-dose, placebo-controlled trial. Clin J Pain 2013, 29, 857–864. DOI:10.1097/AJP.0b013e31827a72d2.
  2. Hu, Y.; Zou, L.; Qi, X.; Lu, Y.; Zhou, X.; Mao, Z.; Chen, X.; Liu, K.; Yang, Y.; Wu, Z.; et al. Subcutaneous botulinum toxin-A injection for treating postherpetic neuralgia. Dermatol Ther 2020, 33, e13181. DOI:10.1111/dth.13181.
  3. Xiao, L.; Mackey, S.; Hui, H.; Xong, D.; Zhang, Q.; Zhang, D. Subcutaneous injection of botulinum toxin a is beneficial in postherpetic neuralgia. Pain Med 2010, 11, 1827–1833. DOI:10.1111/j.1526-4637.2010.01003.x.
  4. Sandre, M. New anatomical insights into preventing brow ptosis with botulinum toxin-A use. Can Dermatol Today 2021, 2, 30–34.

Round 2

Reviewer 1 Report

Comments and Suggestions for Authors

Dear Authors,

Thank you for addressing and answering all the comments and revising your submission. It is now suitable for publication.

Author Response

Dear Authors,

Thank you for addressing and answering all the comments and revising your submission. It is now suitable for publication.

-> Thank you for your valuable suggestion. We have revised the text to clarify and emphasize the novelty of this case report

Reviewer 2 Report

Comments and Suggestions for Authors

All comments and suggestions have been adequately addressed

Author Response

All comments and suggestions have been adequately addressed

  • Thank you for the valuable time taken to review. Thank you for your kind words.
